# Morphology-Controlled Synthesis of V_1.11_S_2_ for Electrocatalytic Hydrogen Evolution Reaction in Acid Media

**DOI:** 10.3390/molecules27228019

**Published:** 2022-11-18

**Authors:** Qiuyue Chen, Siqi Tian, Xiaonan Liu, Xuguang An, Jingxian Zhang, Longhan Xu, Weitang Yao, Qingquan Kong

**Affiliations:** 1College of Chemical Engineering, Sichuan University of Science & Engineering, Zigong 643000, China; 2School of Mechanical Engineering, Chengdu University, Chengdu 610106, China; 3School of Chemical Engineering, Sichuan University, Chengdu 610065, China

**Keywords:** hydrogen evolution reaction, V_1.11_S_2_, morphology, hydrothermal synthesis

## Abstract

High-performance low-cost catalysts are in high demand for the hydrogen evolution reaction (HER). In the present study, we reported that V_1.11_S_2_ materials with flower-like, flake-like, and porous morphologies were successfully synthesized by hydrothermal synthesis and subsequent calcination. The effects of morphology on hydrogen evolution performance were studied. Results show that flower-like V_1.11_S_2_ exhibits the best electrocatalytic activity for HER, achieving both high activity and preferable stability in 0.5 M H_2_SO_4_ solution. The main reason can be ascribed to the abundance of catalytically active sites and low charge transfer resistance.

## 1. Introduction

Nowadays, the development of green and sustainable energy has become an important research topic due to the depletion of fossil fuels, as well as increasing serious environmental issues [1,2]. Therefore, it is urgent to develop renewable and clean alternatives. Electrochemical water splitting is considered to be a clean and sustainable way to produce hydrogen fuel [3]. Moreover, the abundance of protons in acid electrolytes facilitates the acceleration of the hydrogen evolution reaction [4]. However, the acidic electrolyte can cause severe chemical corrosion of electrolyzers, which limits the use of non-platinum group metals or their compounds as catalysts [5]. In particular, the high cost and insufficient reserves of precious metals have greatly restricted their large-scale commercial applications [6,7]. Therefore, a lot of research has been focused on exploring low-cost electrocatalysts [8,9,10,11,12].

Among the various hydrogen evolution reaction (HER) catalysts, transition metal chalcogenides (TMDs) have made tremendous progress due to their high catalytic activity toward the HER, as well as their low-cost [13,14]. MoS_2_ is one of the most excellent electrocatalytic materials among transition metal sulfides, and the catalytic activity and mechanism of MoS_2_ for HER have been widely understood [15,16,17,18]. MoS_2_ with a two-dimensional (2D) layered structure is known to contain both active edge sites and chemically inert basal plane. Lots of work has been conducted to improve the activity by increasing the edge sites of MoS_2_ and/or exploiting the inert basal plane to create additional active sites [19,20]. Hexagonal 1T-phase VS_2_ (1T-VS_2_) as a group TMDs is a promising HER electrocatalyst. The structure of 1T-VS_2_ is similar to that of MoS_2_, which is assembled by stacked S-V-S monolayers via weak van der Waals interaction, which also has excellent structural stability. For the first time, Pan demonstrated by density functional theory calculation that the catalytic performance of single-layer VS_2_ is equivalent to that of Pt at low hydrogen coverage [21]. Zhang and his colleagues further explained the role of intrinsic point defects in HER activity of monolayer VS_2_ catalyst [22]. After that, Liang et al. developed a facile hydrothermal calcination method to synthesize self-supported VS_2_ on carbon paper, which shows excellent HER properties [23]. Qu and his colleagues also prepared VS_2_ with flower-like morphology, obtaining superior HER performance in acid solution [24].

V_1.11_S_2_ phase is one of nonstoichiometric 1T-V_1_ + _X_S_2_ (0 < X< 0.17) with V atoms in the interstitial site between adjacent layers (X is the concentration of V atoms) [25,26]. Both theoretical and experimental results indicate the excellent HER activity of self-intercalated V_1.11_S_2_, which shows a much faster proton/electron adsorption and hydrogen release process than the VS_2_ [26]. Despite these advances, there were few reports focused on the morphology-controlled synthesis of V_1.11_S_2_, as well as their effect on HER performance. It is well known that electrocatalytic activities are highly reliant on the catalyst morphology, which is given more edge sites and lowly coordinated surface atoms that often determine the catalytic performance [27].

Herein, different morphologies of V_1.11_S_2_ were synthesized by a simple hydrothermal synthesis and subsequent calcination (Figure 1). The electrochemical catalytic properties of the resultant V_1.11_S_2_ materials were systematically investigated.

## 2. Results and Discussion

Figure 2 shows the XRD patterns of the obtained V_1.11_S_2_ materials. All the diffraction peaks can be assigned to the V_1.11_S_2_ (33–1445) phase without discernible impurities. It was found that both V_1.11_S_2_-1 and V_1.11_S_2_-2 have a well-crystalline phenomenon. It can be seen from Figure 2 and Appendix A that the XRD diffraction peaks before and after calcination are quite different, which is mainly due to the transformation of VS_4_ and VS_2_ to V_1.11_S_2_ at high-temperature conditions [28].

Figure 3 shows the FE-SEM images of V_1.11_S_2_ materials. Figure 3a,b display that flower-like V_1.11_S_2_ is stacked by a large number of V_1.11_S_2_ nanoplates in different directions. It is worth noting that the average radius of a single V_1.11_S_2_ nanoflower is about 10 μm. Moreover, the formation of flowerlike V_1.11_S_2_ probably involves two steps [29]. First, in a weak alkaline environment, the −SH functional group produced by C_2_H_5_NS reacts with the precursor of V to form V-S intermediate complexes followed by decomposing to shape VSx (x = 2, 4) nuclei for further growth. Then, VSx nanoplates are transformed at high temperatures into V_1.11_S_2_ nanosheets, which are stacked together to form flower-like structures. Irregular flake V_1.11_S_2_ prepared by process B is shown in Figure 3c and Appendix A. It can be seen from Appendix A that the precursor obtained in process B is closely stacked by nanosheets, which are dispersed and smaller after calcination. When the solvent change to ethanol, a porous structure can be observed (Figure 3d). Compared with the powder before calcination shown in Appendix A, the morphology changes dramatically, which is mainly due to the slight solubility of NaVO_3_ in ethanol solution and the calcination procedure [30]. These results suggest the morphology can be easily controlled by altering the hydrothermal solvent and the source of vanadium. Transmission electron microscope (TEM) measurements were performed to analyze the physical structure of the V_1.11_S_2_-1. As depicted in Figure 4a, flower-like V_1.11_S_2_-1 can be exfoliated into a nanosheet structure under long−time ice bath ultrasound. The high-magnification TEM image shown in Figure 4b exhibits the periodic lattice fringe pattern, and the inter-planar spacing was measured to be 0.163 nm, which agrees with that of the (110) facet of V_1.11_S_2_ (Figure 4b). The corresponding selected area electron diffraction (SAED) pattern also confirmed the crystal structure of the V_1.11_S_2_ phase (inset in Figure 4b). Moreover, the EDS pattern in Appendix A also reveals that V and S can be detected.

To detect the surface chemical state and element composition, X-ray photoelectron spectroscopy (XPS) analysis was performed on V_1.11_S_2_ with different morphologies. Investigation of the XPS spectrum clearly shows the presence of V and S (Appendix A). The V 2p spectra can be fitted with two sets of doublet peaks (Figure 5a), and the spectrum of V_1.11_S_2_(V 2p) shows two additional broad peaks at a lower binding energy of 513.4, 516.3, 520.9, and 523.8 eV, which can be respectively assigned to V^2+^2p_3/2_, V^4+^2p_3/2_, V^2+^2p_1/2_, and V^4+^2p_1/2_ [26]. The peak fitting analysis of S 2p (Figure 5b) confirms the presence of S^2−^ with two peaks located at 160.8 and 162 eV that can be assigned to S2p_3/2_ and S2p_1/2_ [26,31]. The combined above-mentioned data indicate that the V_1.11_S_2_ materials with different morphologies have been successfully prepared.

The electrocatalytic HER activities of V_1.11_S_2_ materials were assessed by linear sweep voltammetry (LSV) using a three-electrode system under 0.5 M H_2_SO_4_ acidic aqueous condition. From Figure 6a, V_1.11_S_2_-1 exhibits the best catalytic performance, achieving a current density of 10 mA cm^−2^ with an overpotential of 252 mV, which is superior to the previously reported vanadium sulfide acidic HER electrocatalysts, such as VS_2_ nanodiscs (420 mV) [32], CFP supported V_1.11_S_2_ (259.7 mV) [33], non−templated VS_2_ (378 mV) [34], and Co-N-doped single-crystal V_3_S_4_ nanoparticles (268 mV) [35]. As illustrated in Figure 6b, the calculated Tafel slopes of V_1.11_S_2_-1, V_1.11_S_2_-2, V_1.11_S_2_-3, and 5 wt.% Pt/C are 71.7 mV dec^−1^, 264.1 mV dec^−1^, 102.5 mV dec^−1^, and 30.6 mV dec^−1^, respectively. It is worth noting that the Tafel slope of 5 wt.% Pt/C is as low as 30.6 mV dec^−1^, which is consistent with previous studies [36,37]. Therefore, V_1.11_S_2_-1 shows lower overpotential and Tafel slope, indicating its high HER activities.

The stability of the catalyst plays an important role in practical application. The stability test of V_1.11_S_2_-1 was also carried out by chronopotentiometry test. From Appendix A, the potential remains stable at the current density of 10 mA cm^−2^. For comparison, the potential of 5 wt.% Pt/C drops dramatically with the extension of test time, which is consistent with previous studies [38,39]. After the chronopotentiometry test, the LSV curves of V_1.11_S_2_-1 show a negligible recession phenomenon (Appendix A), suggesting that the catalyst maintains a highly stable catalytic performance. In brief, the above electrochemical test results confirm the flower-like V_1.11_S_2_ material has superior electrochemical activity and stability for HER.

According to previous studies, the catalytic active H−adsorption site of the V_1.11_S_2_ catalyst is S in the outermost layer [24,40]. Generally, the electrocatalytic activity is highly dependent on the catalyst morphology with more active sites. In order to further clarify the origination of excellent HER performance for V_1.11_S_2_ materials, both the electrochemical surface area (ECSA) of the samples were tested. The corresponding current in the applied potential window of 0.06–0.16 V vs. the reversible hydrogen electrode (RHE) should be originated from the charging of the double-layer, and the calculated capacitance (C_dl_) should be proportional to the ECSA [41]. As shown in Figure 7 and the corresponding cyclic voltammograms in Appendix A, V_1.11_S_2_-1 has higher electric double-layer capacitance (3.4 mF cm^−2^) than V_1.11_S_2_-2 (0.45 mF cm^−2^) and V_1.11_S_2_-2 (1.9 mF cm^−2^). Moreover, the fitting value R−Squares is listed in Appendix A, suggesting V_1.11_S_2_-1 has a larger surface area with more exposed active sites. This may be one of the reasons for its high HER performance.

EIS measurements were performed to examine the kinetic differences between V_1.11_S_2_ in different morphologies during the electrochemical process [42]. As shown in the illustration in Figure 8, the semicircle in the Nyquist plots was fitted by using the Randles equivalent circuit, in which Rs represents the equivalent series resistance, Rct_1_ represents the charge transfer resistance of the electrode, and CPE represents the constant phase element [43,44]. It is worth noting that the charge transfer resistance (Rct_1_) is related to the electrocatalytic kinetics, and a lower value corresponds to a faster reaction rate, which can be quantified from the diameter of the semicircle in the low-frequency zone [45]. Table 1 demonstrates the changing trend of the Rct_1_ value for V_1.11_S_2_ nanomaterials with different morphologies, V_1.11_S_2_-1 (49.54 Ω) < V_1.11_S_2_-3 (60.8 Ω) < V_1.11_S_2_-2 (114.3 Ω), indicating that V_1.11_S_2_-1 has better conductivity. Overall, we can conclude that the enhanced catalytic HER activity of flower-like V_1.11_S_2_ compared to the other two structures can be accountable for both the abundant catalytically active sites and preferable low charge transfer resistance.

## 3. Experimental Section

### 3.1. Materials

Ammonium vanadate (NH_4_VO_3_), sodium orthovanadate (Na_3_VO_4_·12H_2_O), sodium metavanadate (NaVO_3_), vanadyl acetylacetonate (C_10_H_14_O_5_V), thioacetamide (CH_3_CSNH_2_), cysteine (C_3_H_7_NO_2_S), ammonia (NH_3_·H_2_O), deionized water (H_2_O), anhydrous ethanol (C₂H₆O), N-methyl pyrrolidone (C_5_H_9_NO). The above chemicals and reagents were purchased from Chengdu Kelong Co., Ltd., Chengdu, China. All reagents were used directly without further purification. The commercial 5 wt.% Pt/C catalyst was purchased from Macleans. Carbon paper (TGP-H-060) was purchased from yilongsheng Energy Technology Co., Ltd., Suzhou, China.

### 3.2. Synthesis of V_1.11_S_2_ Materials

The schematic for the synthesis of V_1.11_S_2_ with different morphology is shown in Figure 1. In process A, 1 mmol NH_4_VO_3_ and 10 mmol CH_3_CSNH_2_ were first dissolved in a solution containing 38 mL deionized water and 2 mL NH_3_·H_2_O, which was stirred for 1 h to form a uniform solution. Afterward, the prepared solution was transferred to a 50 mL Teflon-lined stainless-steel autoclave and maintained at 160 °C for 24 h. After natural cooling to room temperature, the black precipitates were collected by centrifugation, washed several times with deionized water and absolute ethanol, and dried under a vacuum for 6 h. In process B, 1 mmol NaVO_3_ was used as a vanadium source, and 15 mmol CH_3_CSNH_2_ as a sulfur source, dissolved in 25 mL deionized water; other reaction conditions were the same. In process C, 4 mmol NaVO_3_, and 24 mmol CH_3_CSNH_2_ were dissolved in 25 mL C₂H₆O, and the reaction temperature was raised to 180 °C. All final precipitates were calcined at 400 °C for 2 h to obtain V_1.11_S_2_. These V_1.11_S_2_ materials obtained by different preparation procedures were labeled as V_1.11_S_2_-1, V_1.11_S_2_-2, and V_1.11_S_2_-3, respectively.

### 3.3. Materials Characterization

The phase constitutes of the obtained samples were characterized by an X-ray diffractometer (XRD, DX-2700B) with Cu Kα radiation. The microstructure of V_1.11_S_2_ with different morphology was examined by a field emission scanning electron microscope (FESEM, FEI Insect F50). The TEM images of V_1.11_S_2_−1 were obtained by high-resolution transmission electron microscopy (TEM, FEI Talos F200S Super). The surface valence states and elemental compositions were analyzed using X-ray photoelectron spectroscopy (XPS, Thermo Fischer, ESCALAB Xi^+^).

### 3.4. Electrochemical Measurements

All electrochemical data were measured by an electrochemical workstation (CHI660E, CH Instrument, Shanghai, China) with a typical three-electrode electrochemical cell in the acidic electrolyte (0.5 M H_2_SO_4_). A graphite rod was used as the counter electrode, and a saturated calomel electrode was used as the reference electrode. The working electrode was prepared as follows: 3 mg catalysts and 50 μL Nafion solution (5 wt.%) were dispersed in 500 μL mixed solvent of deionized water−isopropanol (volume ratio of 3:1), then sonication to form a homogeneous ink solution. Ink with a volume of 15 μL was loaded onto the carbon paper (0.25 cm^−2^) electrode and dried at ambient temperature. A loading density of about 0.343 mg cm^−2^ was obtained. Linear sweep voltammograms (LSV) were measured from 0.10 to −0.90 V (vs. RHE) at a scan rate of 5 mV s^−1^. Electrochemical impedance spectroscopy (EIS) was conducted under −238 mV (vs. RHE) over a frequency range from 100 kHz to 0.01 Hz with a 5mV amplitude potential. The cyclic voltammograms (CV) measurements at various scan rates from 20, 40, 60, 80, and 100 mV s^−1^ were performed in the potential range of 0.06~0.16 V (vs. RHE) for the electrochemical double−layer capacitance (C_dl_) estimation. It should be noted that no iR compensation was applied to our testing data. According to $E_{\mathrm{corrected}}=E_{\mathrm{measured}}-i\times R_{s}$, all potentials are corrected by iR, where i is the test current and Rs is the equivalent series resistance, which is determined by the Nyquist plots fitting.

## 4. Conclusions

In conclusion, V_1.11_S_2_ was successfully synthesized by hydrothermal synthesis and subsequent calcination. The morphology can be easily controlled by altering the hydrothermal solvent and the source of vanadium. The electrocatalysis results show that the flower−like V_1.11_S_2_ has the best catalytic activity, which can be ascribed to abundant catalytically active sites and preferable low charge transfer resistance. This research provides us with a strong basis for the morphology dependent of V_1.11_S_2_ materials towards HER. Further works will be performed to enhance the intrinsic activity of V_1.11_S_2_ and/or supported on conductive substrates such as carbon cloth and metal foam.

## Figures and Tables

**Figure 1 molecules-27-08019-f001:**
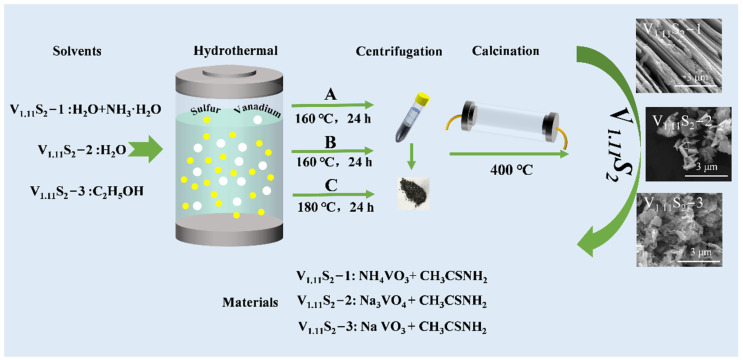
Schematic for the synthesis of V_1.11_S_2_ with different morphology.

**Figure 2 molecules-27-08019-f002:**
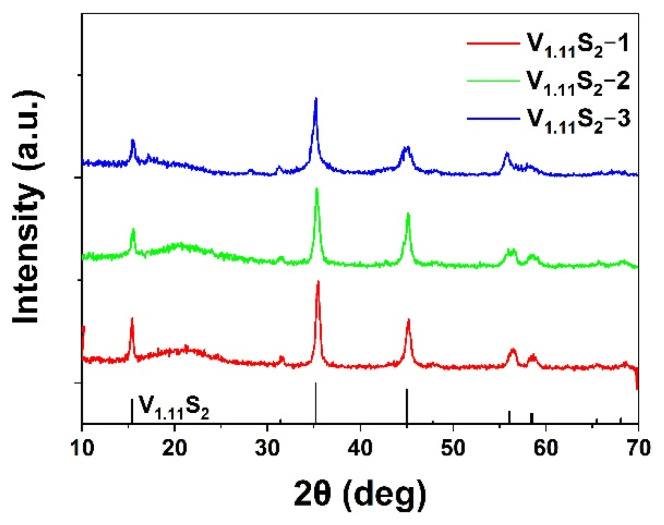
The XRD patterns of the as-annealed V_1.11_S_2_ materials.

**Figure 3 molecules-27-08019-f003:**
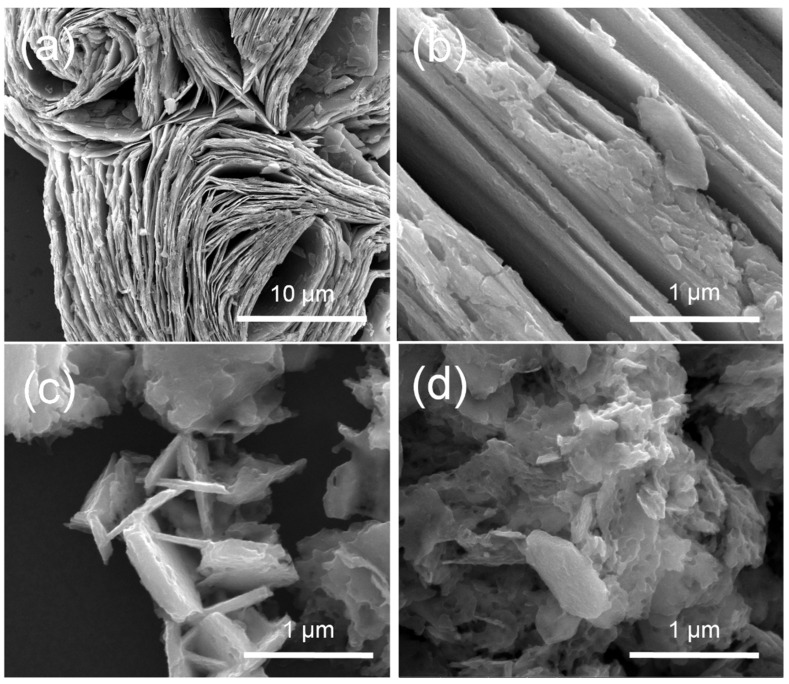
FE-SEM images of V_1.11_S_2_ materials. (**a**,**b**) V_1.11_S_2_-1, (**c**) V_1.11_S_2_-2, (**d**) V_1.11_S_2_-3.

**Figure 4 molecules-27-08019-f004:**
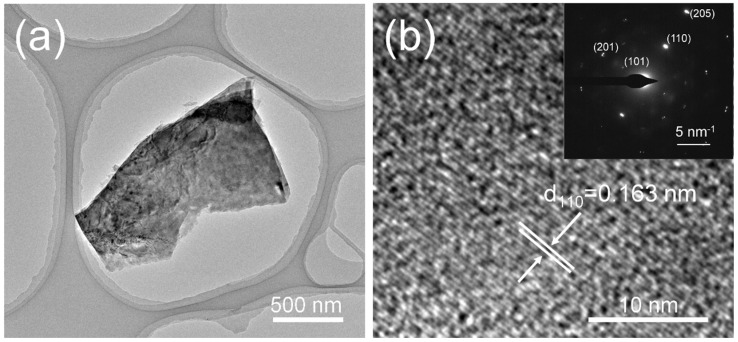
(**a**) TEM images, and (**b**) high-magnification TEM images of the V_1.11_S_2_-1 materials. The inset in Figure (**b**) is the corresponding SAED pattern.

**Figure 5 molecules-27-08019-f005:**
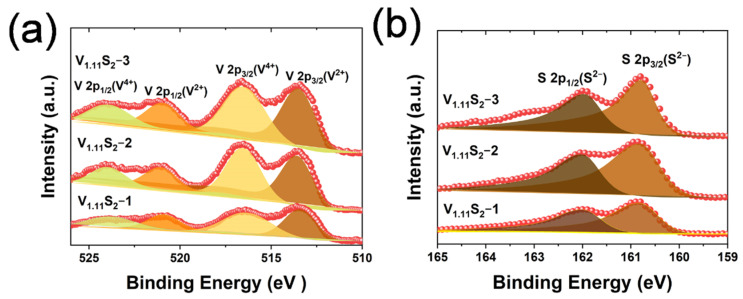
XPS high-resolution spectra of (**a**) V 2p and (**b**) S 2p for the V_1.11_S_2_-1, V_1.11_S_2_-2, and V_1.11_S_2_-3 materials.

**Figure 6 molecules-27-08019-f006:**
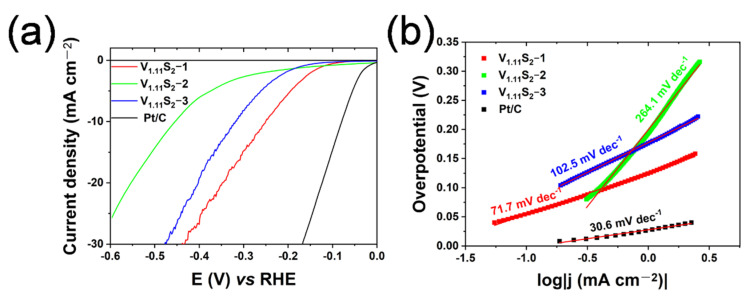
Electrochemical property of V_1.11_S_2_ materials for HER in 0.5 M H_2_SO_4_. (**a**) The iR-corrected polarization curves; (**b**) Tafel plots of V_1.11_S_2_-1, V_1.11_S_2_-2, V_1.11_S_2_-3, and 5 wt.% Pt/C.

**Figure 7 molecules-27-08019-f007:**
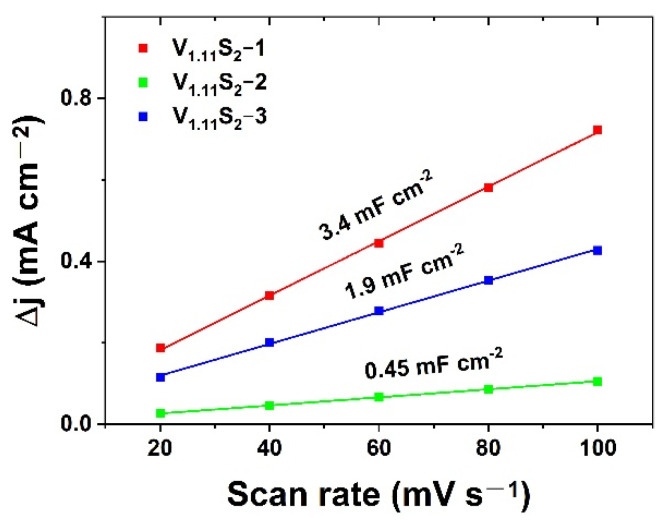
Estimation of Cdl by liner fitting the differences in current density variation (Ja-Jc) at 0.15 V (vs. SCE) as a function of scan rate.

**Figure 8 molecules-27-08019-f008:**
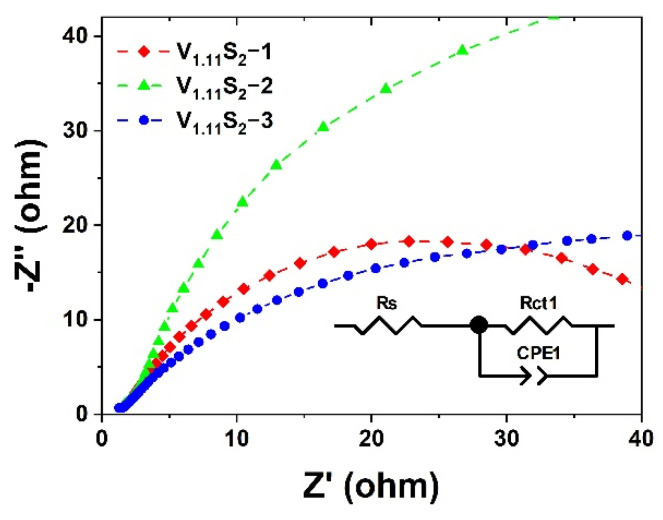
Nyquist plots for V_1.11_S_2_ materials without iR compensation.

**Table 1 molecules-27-08019-t001:** The electrochemical equivalent circuit (EEC) parameters of V_1.11_S_2_ nanomaterials with different morphologies were obtained by fitting the Nyquist diagram.

Catalyst	V_1.11_S_2_-1	V_1.11_S_2_-2	V_1.11_S_2_-3
Rs(Ω)	0.36 ± 0.023	0.34 ± 0.014	0.49 ± 0.021
Rct_1_ (Ω)	49.54 ± 0.48	114.3 ± 3.27	60.8 ± 1.08
CPE_2_-T(F)	2.03 × 10^−3^ ± 7.77 × 10^−6^	1.84 × 10^−4^ ± 1.32 × 10^−5^	2.15 × 10^−3^ ± 4.84 × 10^−5^
CPE_2_-P(F)	0.82 ± 5.33 × 10^−3^	0.90 ± 0.012	0.69 ± 3.89 × 10^−3^

## Data Availability

The data presented in this study are available on request from the corresponding author. The data are not publicly available due to restrictions eg privacy or ethical.

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
