# Peer review of "Morphology-Controlled Synthesis of V1.11S2 for Electrocatalytic Hydrogen Evolution Reaction in Acid Media"

_molecules, 2022, doi:10.3390/molecules27228019_

Round 1
Reviewer 1 Report
The authors analyse the effect of morphology on the catalyst performance. I think that results are clearly reported . In my opinion the quality of the paper should be improved by performing stability experiments (figure S7) not only at 10mA cm-2 but also at higher currents if possible, e. g. 50 and 100mA cm-2.
Author Response
Thank you very much for sending us the reviewing report about our manuscript (Manuscript ID: molecules-2009453).
Thanks for your valuable suggestions. Our report aims to answer critical questions regarding the effect of morphology on V1.11S2 electrocatalytic hydrogen evolution in an acidic medium. The overpotential required to achieve a current density of 10 mA cm is an important performance indicator for HER catalysts, as it is approximately 10% of the current density of a highly efficient solar fuel conversion device. (Chemical reviews, 2010, 110, 6446-6473. Journal of Power Sources, 2017, 347, 193-200. Journal of the American Chemical Society, 2015, 137 4347-4357.) As shown in Fig. S6, the potential remains stable at the current density of 10 mA cm-2, which demonstrates that V1.11S2-1 has good stability. At this point, it is not necessary to test the stability under high current density. We will take your suggestion in subsequent experiments.
Reviewer 2 Report
It is well known that electrocatalytic activities are highly reliant on the cat- 58 alyst morphology, which is given more edge sites and lowly coordinated surface atoms 59 that often determine the catalytic performance.
The influence of the electrocatalyst morphology on the electrocatalytic performance is an interesting topic. In this work an interesting material of V11.1S2 was obtained following three routes. Nevertheless, there are several major issues that should be addressed before the publication of the manuscript.
1. I am not convinced if the activity of sample 1 is the highest because of material morphology. The XRD diffractograms suggest that sample 1 has the lowest crystallinity. Maybe we can attribute the highest activity to the lowest crystallinity of the sample? Moreover, the low-magnification SEM images of samples 2 and 3 (with the magnification like in Fig. 3a) should be provided to compare with the flower-like morphology of sample 1.
2. Can the authors provide an explanation why the flower-like structure was formed in the conditions of synthesis of sample 1?
3. Procedure B should be described better. How many mmol NaVO3, how many CH3CSNH2
4. Can the authors provide the JCPDS number for the V11.1S2 refence?
5. How do the Authors know that 0.163 nm interplanar spacing can be attributed to C1.11S2? Was it calculated or found in some reference (reference should be provided).
6. Why are the high-resolution XPS spectra of samples 2 and 3 moved to supplementary information file? For comparison it will be better to present all the spectra in the same figure, as in the case of XRD diffractograms.
7. The electrocatalytic activity of all 3 obtained samples is rather low compared to the literature on the TMDs for HER in acidic medium. Why?
8. English language requires major revision – i.e. sentence: “Since abundant protons in acidic electrolytes are conducive to accelerating hydrogen evolution reaction” or “Pan et al. firstly reported that the catalytic HER performance of single-layer VS2 is comparable performance to that of Pt at low hydrogen coverage by density functional theory calculation” or “Generally, the electrocatalytic activity is highly dependent on the catalyst morphology, which has more active sites.”
Author Response
Dear Reviewers,
Thank you very much for sending us the reviewing report about our manuscript (Manuscript ID: molecules-2009453).
- Thanks for your suggestion. The electrocatalysis results show that sample 1 has the best catalytic activity, which can be ascribed to abundant catalytically active sites and preferable low charge transfer resistance. Additionally, we have provided the low-magnification SEM images (Fig.S3) of samples 2 and 3 for comparison with the flower morphology of sample 1.
- Thanks for your valuable suggestions. The formation of flowerlike V1.11S2 probably involves two steps. First, in a weak alkaline environment, the −SH functional group produced by C2H5NS reacts with the precursor of V to form V−S intermediate complexes, followed by decomposing to shape VSx (x=2, 4) nuclei for further growth. Then, VSx nanoplates are transformed at high temperatures into V1.11S2 nanosheets, which are stacked together to form flower-like structures.
- We apologize for our negligence. We have added the amount of NaVO3 and CH3CSNH2 in the experimental part.
- Thanks for your suggestion. We have supplemented the JCPDS of V1.11S2 (33-1445) in the manuscript.
- As shown in Fig.2, all the diffraction peaks of V1.11S2-1 can be assigned to the V1.11S2 (33-1445) phase without discernible impurities. Moreover, the crystal plane spacing measured by HR-TEM results can be attributed to the (110) crystal plane (0.1639nm) of V1.11S2.
- Thanks for your valuable suggestions. We have supplied the high-resolution XPS spectra of samples 1, 2, and 3 in Fig.5.
- According to previous studies (Nano Energy 2018. 51. 786-792, Electrochim Acta 2019. 300. 208-216), the H adsorption center of transition metal sulfides is S with an edge. The lower catalytic activity of V1.11S2 materials compared to TMDs can be ascribed to insufficient S at the edge. In subsequent studies, we will further improve the electrochemical properties of V1.11S2 through doping or compounding strategies.
- Thanks for your valuable suggestions. We carefully checked and revised the English language in the manuscript.
Round 2
Reviewer 1 Report
I suggest to the authors the readings of two recent papers
1. https://doi.org/10.1007/s43979-022-00022-8
2. https://doi.org/10.1016/j.jpowsour.2020.228619
where PEM electrolyzers work at currents exceeding 1 A cm-2. However I agree with their preliminary investigation and hope they will also find stability at higher current density.
Author Response
Dear Reviewers,
Thank you very much for sending us the reviewing report about our manuscript (Manuscript ID: molecules-2009453).
- Thanks for your valuable suggestions, which have important guiding significance for our research. And we will take your suggestion in subsequent experiments.
Reviewer 2 Report
Thank you for the response but still the sentence: "Moreover, the abundance of protons in acid electrolytes facilitates the acceleration of hydrogen precipitation reaction" is incorrect. The discussed reaction is a hydrogen evolution reaction, not a precipitation reaction.
Author Response
Dear Reviewers,
Thank you very much for sending us the reviewing report about our manuscript (Manuscript ID: molecules-2009453).
Thanks for your valuable suggestions. We apologize for our negligence. We have carefully revised the English language in the manuscript.